# Maximum Total Correlation Reinforcement Learning

## Abstract

Simplicity is a powerful inductive bias. In reinforcement learning, regularization is used for simpler policies, data augmentation for simpler representations, and sparse reward functions for simpler objectives, all that, with the underlying motivation to increase generalizability and robustness by focusing on the essentials. Supplementary to these techniques, we investigate how to promote simple *behavior* throughout the duration of the episode. To that end, we introduce a modification of the reinforcement learning problem, that additionally maximizes the total correlation within the induced trajectories. We propose a practical algorithm that optimizes all models, including policy and state representation, based on a lower bound approximation. In simulated robot locomotion environments, our method naturally generates policies that induce periodic and compressible trajectories, and that exhibit superior robustness to noise and changes in dynamics compared to baseline methods, while also improving performance in the original tasks.

## 1 Introduction

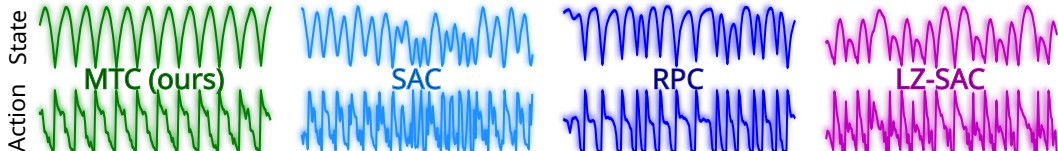

Figure 1: Maximizing the total correlation within trajectories results in more consistent behavior. As shown in our experiments[1], this consistency increases robustness to noise and frictions mismatch.

Reinforcement learning (RL) is currently the preferred approach for many challenging, practical control problems, as it can learn complex neural network policies that effectively tackle the given task. For example, in robotics, reinforcement learning is widely used to learn visuomotor policies for quadrupedal and bipedal locomotion (Lee et al., 2020a; Radosavovic et al., 2024). However, since RL is a learning-based method, it is prone to picking up spurious correlations between high-dimensional sensory inputs and desired actions, which can lead to brittle policies, that fail under slight, natural variations in the state. An important countermeasure involves training policies using domain randomization, in particular in the sim-to-real setting, where the policy is learned in several varying simulation environments. Yet, even in such data-intense settings, it remains unclear whether we can obtain a sufficiently diverse training distribution to learn policies that transfer to more complex real-world scenarios, such as those involving robot controllers that need to interact with humans.

Consequently, there is a growing interest in exploring additional techniques that add inductive biases to obtain simpler, less brittle policies, for example, by limiting the amount of state information used by the policy (Goyal et al., 2018; Igl et al., 2019; Lu et al., 2020), or the predictive information within learned representations (Lee et al., 2020b). Such information-theoretic biases have already been extended to sequences, to account for the sequential nature of reinforcement learning. Namely, RPC (Eysenbach et al., 2021) aims to learn better representations by limiting the information between state-sequences and embedding-sequences, and LZ-SAC (Saanum et al., 2023) improves

---

[1]The code can be found in the supplementary.

the predictability of the next action given the history of actions. However, these formulations only focus on specific aspects of the behavior—either state-consistency or action-consistency—without considering the complete behavior in terms of state-action trajectory.

In this work, we propose a novel inductive bias that operates on the level of trajectories. Specifically, we aim to learn policies that produce simple, consistent, and therefore compressible trajectories. Our hypothesis is that behavior that avoids unnecessary variations tends to be less brittle in general, thereby increasing its robustness to noise and changes in the dynamics. We introduce this inductive bias by means of the additional objective of maximizing the total correlation within the trajectory produced by the agent. This total correlation corresponds to the amount of information that we can save by using a joint encoding of all (latent) states and actions within trajectories, compared to compressing all time steps independently. By maximizing total correlation, the agent is encouraged to produce compressible trajectories, such as periodic and symmetric gaits.

The main contributions of our work are as follows. We introduce the maximum total correlation reinforcement learning problem (MTC-RL), which extends the typical RL formulation with an additional objective of maximizing trajectory total correlation. We derive a lower-bound approximation of the total correlation and use it to propose a practical algorithm for MTC-RL, based on soft-actor critic (Haarnoja et al., 2018). Our algorithm features an adaptation scheme to automatically adapt the coefficient of the total correlation objective by treating it as the Lagrangian multiplier of a constrained optimization problem. We empirically evaluate our algorithm on eight tasks from the DMC control suite (Tassa et al., 2018) and show that the learned policies induce more periodic and better compressible trajectories than baseline methods (Eysenbach et al., 2021; Saanum et al., 2023), leading to an improve in performance, as well as robustness to observation noise, action noise, and changes in the system frictions.

## 2 RELATED WORK

Information theory provides effective tools to solve problems in RL (Memmel et al., 2022; Peters et al., 2010; Ma et al., 2023; Chakraborty et al., 2023; Tishby & Zaslavsky, 2015), such as representation learning (Oord et al., 2018), robustness (Haarnoja et al., 2018), and generalization (Goyal et al., 2018; 2017). Motivated by the InfoMax principle (Bell & Sejnowski, 1995), some previous RL methods preserve mutual information to extract useful representations from observations, and have achieved improvement in terms of performance and robustness on downstream tasks (Kim et al., 2019; Laskin et al., 2020; Mazoure et al., 2020; Rakelly et al., 2021; Dunion et al., 2024). These methods usually maximize mutual information in single transitions. In contrast, our approach maximizes the total interdependencies within the trajectories of an agent. Moreover, instead of using separated objectives to optimize policies and representations, we use a unified objective to optimize policy and representations with respect to the consistency within the resulting trajectories.

Total correlation is a fundamental concept in information theory to qualify the statistical dependency among multiple random variables (Watanabe, 1960). Previous methods have shown that total correlation is an effective tool to enhance machine learning models in many tasks, such as disentangled representation learning (Steeg, 2017; Gao et al., 2019) or structure discovery (Ver Steeg & Galstyan, 2014). Our work extends these results to the RL setting by observing that the agent can actively change its behavior to maximize consistency within state and action sequences. Our method is also related to previous methods that endow RL agents with robust behavior (Tessler et al., 2019; Tanabe et al., 2022; Reddi et al., 2023; Zhang et al., 2020). While these methods have proposed purpose-designed methods to achieve robustness benefits, we focus on demonstrating that maximizing the total correlation is a simple and effective task-independent solution for improving robustness.

The principle of simplicity has garnered substantial attention in constructing learning agents (Chater & Vitányi, 2003; Tishby & Zaslavsky, 2015; Grau-Moya et al., 2018; Igl et al., 2019; Goyal et al., 2018; Tishby & Polani, 2010; Leibfried & Grau-Moya, 2020). Some previous works induce simple policies by imposing temporal consistency in actions. For example, Saanum et al. (2023) propose to capture the temporal consistency in action sequences and induce simple behaviors by incorporating the preference for consistent actions into the reward function. Another class of methods enforces temporal consistency in latent representations of states to obtain policies that produce simple behaviors. For instance, RPC (Eysenbach et al., 2021) learns policies that visit states whose representations are temporally consistent in individual transitions, by minimizing the mutual information between a

sequence of observations and a sequence of their representations. In contrast, our total correlation objective maximizes the consistency among sequences of state representations and actions. This difference, which corresponds to learning dynamic models that predict the future from a history of actions and states, allows the agent to achieve consistent behavior throughout whole trajectories.

# 3 PRELIMINARIES AND NOTATIONS

In this section, we provide a brief overview of the information theory background and the reinforcement learning setting, and introduce the notation used throughout the paper.

## 3.1 INFORMATION THEORY BACKGROUND

Mutual information (MI) is a commonly used statistical dependency measurement in machine learning (Alemi et al., 2017). Given two random variables $x_1$ and $x_2$, their mutual information is defined as:

$$\mathcal{I}(x_1; x_2) = \mathbb{E}_{x_1, x_2} \left[ \log \frac{p(x_1, x_2)}{p(x_1)p(x_2)} \right].$$

Total correlation, or multi-information, generalizes mutual information to more than two random variables (Watanabe, 1960; Studenỳ & Vejnarová, 1998). The total correlation $\mathcal{C}(x_1; x_2; \ldots; x_n)$ of $n$ random variables $x_i$, is defined as the Kullback-Leibler (KL) divergence between the joint distribution and the product of their marginals,

$$\mathcal{C}(x_1; x_2; \ldots; x_n) = \mathbb{E}_{x_1, x_2, \ldots, x_n} \left[ \log \frac{p(x_1, x_2, \ldots, x_n)}{\prod_{i=1}^{n} p(x_i)} \right].$$

This KL divergence corresponds to the expected amount of information (measured in nats), that we can save when transmitting the sequence $(x_1, \ldots, x_n)$ using a code that is optimized with respect to the complete sequence, compared to independently encoding each random variable $x_i$.

## 3.2 MARKOV DECISION PROCESS

We formulate the maximum total correlation reinforcement learning problem in a finite horizon Markov decision process (MDP), denoted by the tuple $\mathcal{M} = (\mathcal{S}, \mathcal{A}, p, r, T)$, where $\mathcal{S}$ is the state space, $\mathcal{A}$ is the action space, $p(s_{t+1}|s_t, a_t)$ is the stochastic dynamic model, $r(s, a)$ is the reward function, and $T$ is the time horizon. At each time step, the agent observes the current state $s_t$ and selects its actions $a_t$ based on its stochastic policy $\pi(a_t|s_t)$ and then receives the reward $r(s_t, a_t)$. The original reinforcement learning objective is to maximize the expected cumulative rewards $\mathbb{E}_\tau \left[ \sum_{t=1}^{T} r_t \right]$ where $\tau = (s_1, a_1, s_2, a_2, \ldots)$ denotes the agent's trajectory. As typically not all state information is relevant for choosing the optimal action, we will assume, without loss of generality, that the policy chooses the action based on a latent variable $z_t \sim f(z_t|s_t)$ using a learned encoder $f$. We refer to the parameters of encoder and policy by $\theta$ and $\phi$, respectively, and we sometimes write $\pi_\phi$ and $f_\theta$ to make this dependency explicit, however, we typically omit the subscript for brevity.

While we use the finite horizon setting for formulating MTC-RL to ensure that the total correlation of trajectories takes finite values, we will transition to the infinite horizon setting in Section 4.3, by letting $T$ go to infinity and introducing a discount factor $\gamma$. In the infinite horizon setting, which underlies the practical implementation used in our experiments, the agent aims to maximize the expected discounted cumulative rewards $\mathbb{E}_\tau \left[ \sum_{t=1}^{\infty} \gamma^t r_t \right]$.

# 4 MAXIMUM TOTAL CORRELATION REINFORCEMENT LEARNING

In this section, we introduce the maximum total correlation reinforcement learning problem, derive a variational lower bound on the total correlation, and use it to formulate an optimization problem that can be solved with existing reinforcement learning methods.

## 4.1 PROBLEM FORMULATION AND MOTIVATION

We want to bias the policy towards producing simpler behavior in order to increase its robustness towards state-, action- or dynamics-perturbations. We quantify the simplicity of the behavior by the total correlation of the complete trajectories induced by the policy, which corresponds to their compressibility in an information-theoretic sense. More specifically, we extend the vanilla reinforcement learning objective by introducing the additional objective of maximizing the total correlation within the trajectory of latent state representations and actions,

$$\max_{\theta,\phi} \quad \mathbb{E}_{\pi_\phi,f_\theta}\left[\left[\sum_{t=1}^{T} r(s_t, a_t)\right] + \alpha \mathcal{C}(z_1; a_1; \dots; a_{T-1}; z_T)\right]. \tag{1}$$

where the hyper-parameter $\alpha$ controls the trade-off between both objectives.

Using the latent representation $z$ rather than the raw states $s$ for the total correlation objective serves two main purposes. Firstly, biasing the policy to actively reduce variability within task-irrelevant state information could result in distractions that mirror the Noisy-TV problem of curiosity-driven exploration methods (Burda et al., 2018), where the agent is attracted to task-irrelevant novelty rather than task-relevant novelty. By restricting our total correlation objective to task-relevant state information, we focus on learning behavior that is consistent only with respect to aspects of the state that actually matter for the task. The second motivation for formulating the total correlation with respect to the learned state representation, is to not only learn more consistent behavior, but also more consistent representations $z$. By penalizing unnecessary variations in the representation, we aim to learn representations that are more robust to irrelevant variations in the state.

## 4.2 A VARIATIONAL BOUND ON TOTAL CORRELATION

The total correlation objective in Eq. 1 can not be decomposed into a sum of step-rewards and involves probability distributions that are typically not available in analytic form. Hence, we replace it with a variational lower bound, using a history-based latent dynamics model $q_\eta(z_{t+1}|z_{1:t}, a_{1:t})$ and a history-based action prediction model $q_\chi(a_t|z_{1:t}, a_{1:t-1})$,

$$\widetilde{\mathcal{C}}(z_1; a_1; \dots; a_{T-1}; z_T) = \mathbb{E}_{\pi,f}\left[\sum_{t=1}^{T-1}\left[\log\frac{q_\eta(z_{t+1}|z_{1:t}, a_{1:t})}{f_\theta(z_{t+1}|s_{t+1})} + \log\frac{q_\chi(a_t|z_{1:t}, a_{1:t-1})}{\pi_\phi(a_t|s_t)}\right]\right] \tag{2}$$

$$\leq \mathcal{C}(z_1; a_1; \dots; a_{T-1}; z_T).$$

Please refer to Appendix A.1 for the derivation. The contribution of a given time step $t$ to the lower bound is large, when the next latent state and the next action can be predicted well based on the history, while accounting for the irreducible uncertainty due to the stochastic encoder $f$ and the policy $\pi$. Hence, this mechanism encourages coherent and consistent trajectories. As shown in our experiments, both state consistency and action consistency are significantly improved when using the lower bound $\widetilde{\mathcal{C}}$ within the MTC-RL objective (see Figure 1), which demonstrates that the lower bound captures important aspects of the total correlation.

## 4.3 A TRACTABLE OPTIMIZATION PROBLEM

By plugging the lower bound $\widetilde{\mathcal{C}}(z_1; a_1; \dots; a_{T-1}; z_T)$ in Eq. 2 into the objective function Eq. 1, we obtain the tractable objective

$$\max_{\theta,\phi,\eta,\chi} \quad \mathbb{E}_{\pi_\phi,f_\theta}\left[\sum_{t=1}^{T-1}\left[r(s_t, a_t) + \alpha \log\frac{q_\eta(z_{t+1}|z_{1:t}, a_{1:t})q_\chi(a_t|z_{1:t}, a_{1:t-1})}{f_\theta(z_{t+1}|s_{t+1})\pi_\phi(a_t|s_t)}\right] + r(s_T, a_T)\right] \tag{3}$$

that we optimize with respect to the parameters of the policy, encoder, and latent dynamics model.

The policy is, thus, optimized with respect to the information-regularized reward function

$$r^*(s_t, a_t, s_{t+1}) = r(s_t, a_t, s_{t+1}) + \alpha\left(\log\frac{q_\eta(z_{t+1}|z_{1:t}, a_{1:t})q_\chi(a_t|z_{1:t}, a_{1:t-1})}{f_\theta(z_{t+1}|s_{t+1})\pi_\phi(a_t|s_t)}\right). \tag{4}$$

The modified reward biases the policy towards states for which the latent representation can be well-predicted based on the history, relative to the uncertainty in the encoder predictions, and towards actions that can well predicted by the action prediction model, relative to the uncertainty of the policy.

The latent history-based dynamics and action prediction models get trained using maximum likelihood, and the encoder and policy get biased towards the history-based predictions, due to the additional objectives of minimizing the KL divergence towards history-based models.

For the practical implementation, we switch to the infinite horizon problem setting by letting $T \to \infty$, and introducing the discount factor $\gamma$, that is, we optimize the final objective function

$$\max_{\theta,\phi,\eta,\chi} \quad \mathbb{E}_{\pi_\phi,f_\theta}\left[\sum_{t=1}^{\infty} \gamma^t r^*(s_t, a_t, s_{t+1})\right]. \tag{5}$$

### 4.4 Maximum Total Correlation Soft Actor Critic

Our total correlation regularized reinforcement learning problem in Eq. 5 can be optimized straightforwardly with existing RL methods. For our experiments we implement MTC on top of soft actor-critic (SAC) (Haarnoja et al., 2018). As an actor-critic method, SAC alternates between estimating the Q function (policy evaluation) and improving the policy with respect to the Q function (policy improvement). SAC considers the maximum entropy RL setting, that is, it has the additional objective of maximizing the entropy of the policy, and therefore, it computes the soft-Q function $Q_{\text{soft}}^\pi(s, a)$ during policy evaluation, which also accounts for the expected future entropy of the policy. For our policy evaluation, we do not need to make any modifications to SAC, besides replacing the original reward function $r(s_t, a_t)$ with the regularized reward $r^*(s_t, a_t, s_{t+1})$. Hence, we also learn the soft-Q function and use common techniques such as target networks (Mnih et al., 2015) and dual Q networks (Fujimoto et al., 2018; Haarnoja et al., 2018).

For policy improvement, however, we also optimize the dynamics model, the action prediction model and the encoder along with policy. While the prediction models are, thus, trained on the replay buffer instead of using on-policy samples, which slightly deviates from the derived update and may increase the gap of our lower bound, this change allows for an easy integration of the total-correlation regularizer for off-policy optimization. Similar to RPC (Eysenbach et al., 2021), we express the soft-Q function in terms of the regularized reward and the soft Q-function of the next time step, to arrive at the following objective,

$$\max_{\theta,\phi,\eta,\chi} \quad \mathbb{E}_{\mathcal{D},\pi_\phi,f_\theta}\left[\alpha\Big(\log \frac{q_\eta(z_{t+1}|z_{1:t}, a_{1:t})q_\chi(a_t|z_{1:t}, a_{1:t-1})}{f_\theta(z_{t+1}|s_{t+1})}\Big) - (\alpha + \beta)\log(\pi_\phi(a_t|s_t))\right.$$
$$\left. + \gamma\Big(Q_{\text{soft}}^\pi(s_{t+1}, a_{t+1}) - (\alpha + \beta)\log(\pi_\phi(a_{t+1}|s_{t+1}))\Big)\right], \tag{6}$$

where $s_{1:t+1}$ and $a_{1:t}$ are sampled from the replay buffer $\mathcal{D}$, $a_{t+1}$ is sampled from the current policy, and all embeddings $z_{1:t+1}$ are sampled from the current encoder. The coefficient $\beta$ corresponds to the weight of the entropy regularizer of SAC.

Furthermore, instead of choosing the hyperparameter $\alpha$ directly, we optimize it with respect to a desired bound $I_p$, by minimizing the dual objective

$$L(\alpha) = \alpha\Big(\log \frac{q_\eta(z_{t+1}|z_{1:t}, a_{1:t})q_\chi(a_t|z_{1:t}, a_{1:t-1})}{f_\theta(z_{t+1}|s_{t+1})\pi_\phi(a_t|s_t)} - I_p\Big). \tag{7}$$

The training procedure of our algorithm is presented in Algorithm. 1. The algorithm proceeds by alternating between collecting new experiences from the environment, and updating the parameters of our model. We use an LSTM (Hochreiter & Schmidhuber, 1997) for the history-based models and limit the maximum length of the history using a hyperparameter $l$.

## 5 Experimental Evaluation

We performed experiments to investigate how our total correlation objective compares to vanilla soft-actor critic (Haarnoja et al., 2018) and the closely related alternative methods RPC (Eysenbach

---

**Algorithm 1:** MTC

---

**Initialize:** policy $\pi_\phi(a_t|z_t)$, Q function $Q_\upsilon(s_t, a_t)$, encoder $f_\theta(z_t|s_t)$, dynamic model $q_\eta(z_{t+1}|z_{1:t}, a_{1:t})$, action prediction model $q_\chi(a_t|z_{1:t}, a_{1:t-1})$, replay buffer $\mathcal{D}$, coefficients $\alpha, \beta$, history $l$, batch size $B$, learning rate $\rho$

**for** *each training step* **do**

   collect experience $(s_t, a_t, r_t, s_{t+1})$ and add it to replay buffer

   **for** *each gradient step* **do**

      Sample a minibatch of transitions from replay buffer: $\{(s_t^i, a_t^i, r_t^i, s_{t+1}^i)_t\}_{i=1}^B \sim \mathcal{D}$

      Compute lower bound: $b \leftarrow \mathbb{E}\left[\log \frac{q_\eta(z_{t+1}|z_{t-l:t})q_\chi(a_t|z_{1:t}, a_{1:t-1})}{f_\theta(z_{t+1}|s_{t+1})\pi_\phi(a_t|z_t)}\right.$         $\triangleright$ Eq. 2

      Compute information-regularized reward: $r^* \leftarrow r + \alpha b$         $\triangleright$ Eq.4

      Update Q function: $\upsilon \leftarrow \upsilon - \rho \hat{\nabla}_\upsilon \mathcal{L}(\upsilon)$         $\triangleright$ Eq.14

      Update policy, encoder, dynamics and action prediction model:

        $\{\phi, \theta, \eta, \chi\} \leftarrow \{\phi, \theta, \eta, \chi\} - \rho \hat{\nabla}_{\{\phi, \theta, \eta, \chi\}} \mathcal{L}(\phi, \theta, \eta, \chi)$         $\triangleright$ Eq. 6

      Update multiplier $\alpha$: $\alpha \leftarrow \alpha - \rho \hat{\nabla}_\alpha \mathcal{L}(\alpha)$         $\triangleright$ Eq. 7

      Update multiplier $\beta$

   **end**

**end**

---

et al., 2021) and LZ-SAC (Saanum et al., 2023) in terms of performance on the original RL objective (Sec. 5.1), robustness to noise and friction mismatch (Sec. 5.2), and consistency of the resulting trajectories (Sec. 5.3). Furthermore, we performed ablations to investigate the effects for different total correlation constraints $I_p$ (Sec. 5.4).

We evaluate our method on eight continuous control tasks from the DeepMind Control (DMC) (Tassa et al., 2018), a commonly used open-source simulated benchmark in RL settings. We build MTC on top of the open source implementation of SAC by Yarats et al. (2021). Whereas the official implementation of LZ-SAC provided by Saanum et al. (2023) also uses this SAC implementation, the original implementation of RPC provided by Eysenbach et al. (2021) is based on the SAC implementation from TF-Agents. To ensure a reliable and fair comparison to RPC, we compare MTC to RPC implemented by its original code (referred to as RPC-Orig in Table. 1) and to our implementation of RPC built on top of the same SAC codebase as MTC and LZ-SAC (referred to as RPC). Please refer to Appendix B for details on the implementations of the different approaches.

Table 1: Scores (means over 20 seeds with 90% confidence interval) achieved by our method and baselines on eight DMC tasks at 1 million environment steps. MTC achieves better or at least comparable asymptotic performance than all baselines. In particular, MTC outperforms LZ-SAC and RPC by a large margin on five tasks.

| Scores | MTC | RPC | RPC-Orig | LZ-SAC | SAC |
|---|---|---|---|---|---|
| Acrobot Swingup | **143 $\pm$ 21** | **132 $\pm$ 31** | 20$\pm$3 | 100$\pm$22 | **154 $\pm$ 29** |
| Hopper Stand | **903$\pm$ 24** | 568 $\pm$ 96 | 476$\pm$ 101 | 593$\pm$ 88 | 683$\pm$ 114 |
| Finger Spin | **983$\pm$ 3** | 869 $\pm$ 19 | 921 $\pm$ 13 | 805$\pm$ 38 | 955$\pm$ 18 |
| Walker Walk | **963$\pm$ 3** | 940 $\pm$ 21 | 951 $\pm$ 2 | 939$\pm$ 26 | **962$\pm$ 7** |
| Cheetah Run | **827$\pm$ 36** | 772$\pm$ 57 | 636 $\pm$ 10 | 787$\pm$ 17 | 811$\pm$ 36 |
| Quadruped Walk | **944$\pm$ 5** | 842 $\pm$ 77 | 598 $\pm$ 108 | 595 $\pm$ 110 | 738$\pm$ 93 |
| Walker Run | 770$\pm$ 14 | **778 $\pm$ 25** | 604$\pm$ 29 | 732$\pm$ 22 | **767$\pm$ 13** |
| Walker Stand | **983$\pm$ 2** | **980 $\pm$ 5** | 971 $\pm$1 | 977$\pm$ 2 | **985$\pm$ 2** |

## 5.1 PERFORMANCE

In our first set of experiments we evaluate the performance on the original reinforcement learning problem. Table. 1 shows the final performance of our method and baselines on eight control tasks from DMC. MTC achieves better average asymptotic performance than baselines on the majority of the tasks. In particular, MTC outperforms SAC on four tasks, Hopper Stand, Finger Spin, Cheetah

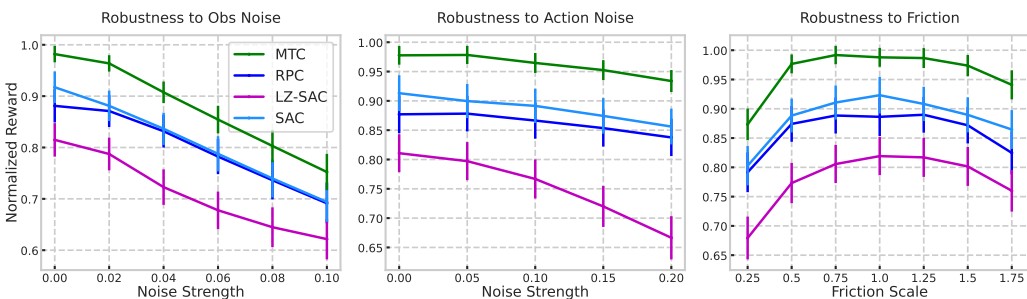

Figure 2: We evaluated the robustness towards observation noise (left), action noise (middle) and friction changes (right) on eight tasks from DMC benchmarks. The plot shows the normalized mean rewards averaged over 20 independent runs and 8 tasks, with error bars representing 90% confidence interval. For each task we normalized the return by the mean return of the best method. Each run includes 30 evaluation trajectories. MTC achieves better aggregated performance than baselines in the presence of perturbations to actions and body frictions, while also obtaining higher mean rewards when observations are perturbed with small Gaussian noise.

Run and Quadruped Walk. These results suggest that inducing simple policies by maximizing the total correlation also benefits policy learning.

## 5.2 ZERO-SHOT ROBUSTNESS

Our main motivation for learning coherent behavior and representations is to improve robustness by focusing on the essentials. Our policies are biased to produce trajectories that have fewer variations, so we expect that they are more robust to unseen disturbances. Hence, we evaluated our method and baselines in terms of zero-shot robustness to observation perturbations, action perturbations, and perturbations to the dynamics.

**Robustness to observation perturbations.** We first investigate how observation perturbations affect policy performance by injecting Gaussian noise into the observations, $s_t \leftarrow s_t + \epsilon$, where noise $\epsilon$ is sampled from a Gaussian distribution, $\epsilon \sim \mathcal{N}(0, \text{diag}(\sigma^2))$ with standard deviation $\sigma$. Using the same tasks as before, we evaluate our method and baselines on a series of noise strength $\sigma \in [0.02, 0.04, 0.06, 0.08, 0.1]$. To compare the robustness across all eight tasks, we normalized the scores by the score achieved by the best method on each task. The aggregated robustness to observation perturbations with different noise strengths is shown for different methods in Figure 2 (left). MTC achieves the best aggregated performance when observations are perturbed with small Gaussian noise.

**Robustness to action perturbations.** As our total correlation objective encourages consistent actions, we also expect an improvement in terms of robustness to action perturbations. Hence, we add Gaussian noise to the actions, $a_t \leftarrow a_t + \epsilon$, where $\epsilon \sim \mathcal{N}(0, \text{diag}(\sigma^2))$ with noise strength $\sigma$. However, please note that we had to clip the values of the noisy actions to be within $[-1, 1]$, due to requirements of the simulator.

Overall, MTC achieves higher average rewards than all baselines even in the presence of strong action perturbations, see Fig. 2 (middle). Notably, our approach outperforms SAC in robustness to action perturbations with different noise strengths, indicating that maximizing trajectory total correlation improves robustness to action perturbations.

**Robustness to frictions.** We also expect simple behavior to be more robust towards deviations between the frictions encountered during testing compared to the frictions used for training. We test the effects of frictions mismatch by scaling the friction of each robot body during evaluation. We evaluate six different scaling factors in each environment, namely $[0.25, 0.5, 0.75, 1.25, 1.5, 1.75]$, and present the aggregated results on eight tasks in Fig. 2 (right). Overall, MTC obtains higher averaged scores than all baselines when body frictions are changed.

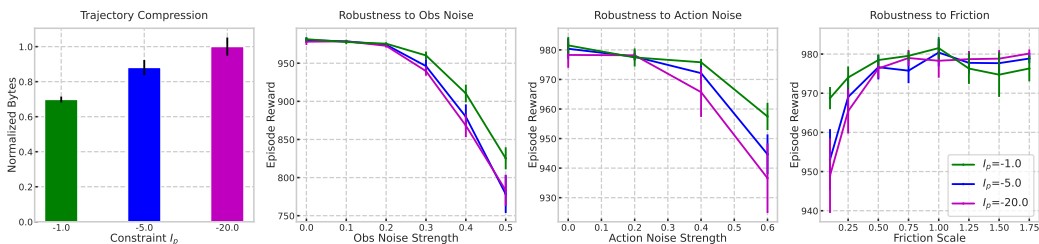

Figure 4: We test the performance of MTC with different constraints $I_p$. All subplots show the mean over 20 independent runs, with error bars representing 90% confidence interval. The robustness to observation noise, action noise and friction changes, and the compression of behaviors are improved while increasing $I_p$.

## 5.3 TRAJECTORY COHERENCE

Arguably, the coherence of a behavior can be most straightforwardly judged by visualizing it. Hence, we generated plots of the state and action trajectories for MTC and all baselines on the Finger Spin task. We already showed the trajectories for the first joint in Fig. 1. The remaining joints are shown in Figure 5 and Figure 6 in Appendix C.1. Based on these visualizations, we argue that MTC produces the simplest and most coherent trajectories, characterized by highly cyclical patterns.

To support this qualitative assessment, inspired by Saanum et al. (2023), we use lossless compression algorithms to quantify the compressibility of trajectories produced by learned policies. We round the collected state-action trajectories to one digit behind the decimal point, save them as .npy-files and compress them using bzip2. Rounding the floating point numbers was necessary to achieve meaningful results because otherwise the highly random insignificant bits would dominate, leading to high variance in the resulting file sizes. Figure 3 shows the normalized average file sizes in bytes among 30 trajectories of 1000 steps for each of the 8 tasks, with error bars representing 90% confidence interval. The normalized file sizes are achieved by dividing the compressed trajectories by the largest compressed trajectory among all methods for each task. Trajectories collected by MTC can be more efficiently compressed than baselines, which suggests that the trajectories produced by

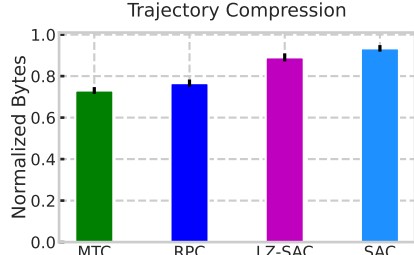

Figure 3: When compressing the state-action trajectories with bzip2, trajectories obtained by MTC result in the smallest filesize in expectation.

our policies show more repetitive, periodic structures to solve tasks. We note, however, that lossless compression algorithms like bzip2 are effective at detecting and compressing repeating patterns in data, but may not always be able to capture more complex or subtle patterns.

## 5.4 HYPERPARAMETER ABLATION

To better investigate the effect of our regularizer, we evaluated the effect of the hyperparameter $I_p$ which is used for optimizing the weight $\alpha$ of the total correlation objective. Increasing the value of $I_p$ results in larger values of $\alpha$ and therefore biases the agent to increase total correlation. We evaluate the effect of $I_p$ with respect to original task performance, compressibility and robustness to state, action and friction perturbations on the Walker Stand task, see Appendix B.10 for more details.

Fig. 4 shows the experimental results. Each subplot shows mean and 90% confidence interval from 30 episodes, averaged over 20 seeds. We observe that tightening the lower bound of our total correlation objective by increasing $I_p$ doesn't hurt the final performance (rewards without perturbations) but significantly decreases the encoding size of trajectories. This suggests that maximizing the lower bound of the total correlation helps induce compressible or structured behaviors. We also find that increasing $I_p$ effectively improves the robustness of learned policies to observation noise, action

noise, and changes in frictions (see Fig. 4). This supports our claim that biasing policies to focus on the essentials helps increase robustness to perturbations.

## 6    DISCUSSIONS AND LIMITATIONS

Our regularizer in Eq. 4 is related to the regularizer of RPC (Eysenbach et al., 2021), but generalizes it by considering the previous trajectory instead of only using the information of the current step $t$, and by also including an action prediction model. These differences enable us to improve temporal consistency within trajectories, which significantly improves the consistency and robustness of the resulting behavior, as shown in our experiments. Furthermore, the regularizer in RPC was derived as the negative of an upper bound on the mutual information between raw and latent state sequences, $I(s_{1:T}; z_{1:T})$, whereas we prove in Appendix A.2, that our objective can not be derived from that perspective. We can, however, derive RPC from our formulation, showing that maximizing total correlation provides an important new perspective on regularization in reinforcement learning that not only results in more coherent and robust behavior, but also deepens our theoretical understanding of related works.

However, our lower bound of the total correlation corresponds to a sum of negated KL divergences, and is therefore always negative. Hence, it is not useful for estimating the actual total correlation, which we know to be positive. While a vacuous bound may not be useful for estimation, it can still be valuable for optimization, as in the case of subtracting a constant offset from the true objective. As demonstrated in our experiments, our lower bound is very effective for producing consistent behavior.

## 7    CONCLUSION AND FUTURE WORK

Auxiliary objectives that create inductive biases towards simpler solutions (regularizers) are commonly, and very successfully, used in machine learning to learn more generalizable and robust solutions. We propose to use the total trajectory correlation as a novel regularizer for reinforcement learning, which acts on the level of the behavior. By directly corresponding to the information-theoretic compressibility of the induced trajectories, the total correlation is arguably the most principled choice to quantify the simplicity of a behavior. As directly maximizing the total correlation is intractable, we derived a variational lower bound and used it to formulate a regularized reinforcement learning problem that can be solved with standard techniques. Compared to similar sequence-based regularizers, total correlation regularization achieved very promising results by producing more coherent behavior that is more robust to state-, action- and dynamics perturbations. Hence, we believe that total trajectory correlation may serve as an important goal post for future reinforcement learning methods. Developing alternate bounds or approximations that better capture the total correlation while maintaining tractability is a promising direction for future research.

## REPRODUCIBILITY STATEMENT

We provide the code with instructions as supplementary material. We describe all implementation details of our method and how we determine hyperparameters in Section B. The implementation details of baselines are also presented in the Section B. Besides, the DMC benchmark we used are open-source, improving the reproducibility of our work.

Moreover, evaluations based on too few seeds can draw misleading conclusions and increase the reproducibility crisis of reinforcement learning. To alleviate this problem, we use 20 seeds for every experiments (performance experiments, robustness experiments, compression experiments, as well as ablations). Besides, we report 90% confidence intervals based on the standard error of the mean, scaled using the critical value from the Student's t-distribution, which ensures the statistical significance of the results.

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

# A    PROOFS

## A.1    DERIVATION DETAILS OF THE LOWER BOUND

In this section, we provide full details about how to derive the lower bound (Eq. 2) from the total correlation definition. We start from the definition of the total correlation and derive a lower bound using a variational approximation $q(z_{1:T}, a_{1:T-1})$ of the trajectory distribution.

$$
\begin{aligned}
\mathcal{C}(z_1; a_1; \ldots; a_{T-1}; z_T) &= \mathbb{E}_{p(z_{1:T}, a_{1:T-1})}\left[\log \frac{p(z_{1:T}, a_{1:T-1})}{\prod_{t=1}^{T} p(z_t) \prod_{t=1}^{T-1} p(a_t)}\right] \\
&= \mathbb{E}_{p(z_{1:T}, a_{1:T-1})}\left[\log \frac{q(z_{1:T}, a_{1:T-1})}{\prod_{t=1}^{T} p(z_t) \prod_{t=1}^{T-1} p(a_t)}\right] \\
&\quad + \mathbb{D}_{\mathrm{KL}}\Big(p(z_{1:T}, a_{1:T-1}) || q(z_{1:T}, a_{1:T-1})\Big) \\
&\geq \mathbb{E}_{p(z_{1:T}, a_{1:T-1})}\left[\log \frac{q(z_{1:T}, a_{1:T-1})}{\prod_{t=1}^{T} p(z_t) \prod_{t=1}^{T-1} p(a_t)}\right].
\end{aligned}
\tag{8}
$$

We parameterize the variational distribution $q(z_{1:T}, a_{1:T-1})$ autoregressively:

$$
q(z_{1:T}, a_{1:T-1}) = p(z_1) q(a_1|z_1) \prod_{t=1}^{T-1} q_\eta(z_{t+1}|z_{1:t}, a_{1:t}) q(a_{t+1}|z_{1:t+1}, a_{1:t}),
\tag{9}
$$

where $q_\eta(z_{t+1}|z_{1:t}, a_{1:t})$ is a history-based dynamics model, $q(a_{t+1}|z_{1:t+1}, a_{1:t})$ a history-based action model.

We plug Eq. 9 into Eq. 8, and then obtain

$$
\begin{aligned}
\mathcal{C}(z_1; a_1; \ldots; a_{T-1}; z_T) &\geq \mathbb{E}_{p(z_{1:T}, a_{1:T-1})}\left[\log \frac{p(z_1) q(a_1|z_1) \prod_{t=1}^{T-1} q_\eta(z_{t+1}|z_{1:t}, a_{1:t}) q(a_{t+1}|z_{1:t+1}, a_{1:t})}{\prod_{t=1}^{T} p(z_t) \prod_{t=1}^{T-1} p(a_t)}\right] \\
&= \mathbb{E}_{p(z_{1:T}, a_{1:T-1})}\left[\log \frac{\prod_{t=1}^{T-1} q_\eta(z_{t+1}|z_{1:t}, a_{1:t})}{\prod_{t=1}^{T-1} p(z_{t+1})}\right] \\
&\quad + \mathbb{E}_{p(z_{1:T}, a_{1:T-1})}\left[\log \frac{\prod_{t=1}^{T-1} q_\chi(a_t|z_{1:t}, a_{1:t-1})}{\prod_{t=1}^{T-1} p(a_t)}\right] \\
&= \mathbb{E}_{p(z_{1:T}, a_{1:T-1})}\left[\sum_{t=1}^{T-1} \log \frac{q_\eta(z_{t+1}|z_{1:t}, a_{1:t})}{p(z_{t+1})}\right] \\
&\quad + \mathbb{E}_{p(z_{1:T}, a_{1:T-1})}\left[\sum_{t=1}^{T-1} \log \frac{q_\chi(a_t|z_{1:t}, a_{1:t-1})}{p(a_t)}\right].
\end{aligned}
\tag{10}
$$

The marginal distributions $p(z_{t+1})$ and $p(a_t)$ are unknown. However, the conditional distributions $f_\theta(z_{t+1}|s_{t+1})$ and $\pi_\phi(a_t|s_t)$ are known and can be substituted while maintaining a lower bound:

$$
\begin{aligned}
&\mathcal{C}(z_1; a_1; \ldots; a_{T-1}; z_T) \geq \\
&\mathbb{E}_{p(z_{1:T}, a_{1:T-1})}\left[\sum_{t=1}^{T-1} \log \frac{q_\eta(z_{t+1}|z_{1:t}, a_{1:t})}{f_\theta(z_{t+1}|s_{t+1})}\right] + \mathbb{E}_{p(s_{1:T}, z_{1:T}, a_{1:T-1})}\left[\sum_{t=1}^{T-1} \log \frac{q_\chi(a_t|z_{1:t}, a_{1:t-1})}{\pi_\phi(a_t|s_t)}\right] \\
&+ \sum_{t=1}^{T-1} \mathbb{E}_{p(s_{t+1})}\left[\mathbb{D}_{\mathrm{KL}}\Big(f_\theta(z_{t+1}|s_{t+1}) \,\|\, p(z_{t+1})\Big)\right] + \sum_{t=1}^{T-1} \mathbb{E}_{p(s_t)}\left[\mathbb{D}_{\mathrm{KL}}\Big(\pi_\phi(a_t|s_t) \,\|\, p(a_t)\Big)\right] \\
&\geq \mathbb{E}_{p(z_{1:T}, a_{1:T-1})}\left[\sum_{t=1}^{T-1} \log \frac{q_\eta(z_{t+1}|z_{1:t}, a_{1:t})}{f_\theta(z_{t+1}|s_{t+1})}\right] + \mathbb{E}_{p(s_{1:T}, z_{1:T}, a_{1:T-1})}\left[\sum_{t=1}^{T-1} \log \frac{q_\chi(a_t|z_{1:t}, a_{1:t-1})}{\pi_\phi(a_t|s_t)}\right]
\end{aligned}
\tag{11}
$$

where the inequality in the last line holds because of the non-negativity of the KL divergence.

### A.2 CONNECTIONS TO $I(s_{1:T}; z_{1:T})$

RPC (Eysenbach et al., 2021) aims to minimize the following upper bound of the mutual information between the state sequence and the latent state sequence,

$$I(s_{1:T}; z_{1:T}) = \mathbb{E}_{p(s_{1:T}, z_{1:T})}\left[\log \frac{p(z_{1:T}|s_{1:T})}{p(z_{1:T})}\right] \leq \mathbb{E}_{p(s_{1:T}, z_{1:T}, a_{1:T})}\left[\log \frac{\prod_{t=1}^{T-1} f(z_{t+1}|s_{t+1})}{\prod_{t=1}^{T-1} q(z_{t+1}|z_t, a_t)}\right]. \tag{12}$$

In contrast to our bound, this bound does not use the history for the dynamics model, and it does not explicitly account for action consistency. Furthermore, we argue that the lower bound (Eq. 12) does not always hold as it was derived by replacing $p(z_{1:T}|s_{1:T})$ by $\prod_{t=1}^{T-1} p(z_{t+1}|s_{t+1})$ (Eysenbach et al., 2021, Appendix C1). These distributions are in general not the same because information about future state observations can decrease uncertainty about the current latent state, and therefore

$$p(z_{t+1}|z_{1:t}, s_{1:T}) \neq p(z_{t+1}|s_{t+1}).$$

We will now show that the latter replacement may invalidate the upper-bound by analyzing the gap,

$$\mathbb{E}\left[\log \frac{p(z_{1:T}|s_{1:T})}{p(z_{1:T})}\right] - \mathbb{E}\left[\log \frac{\prod_{t=1}^{T-1} f(z_{t+1}|s_{t+1})}{\prod_{t=1}^{T-1} q(z_{t+1}|z_t, a_t)}\right]$$

$$= \underbrace{\mathbb{E}_{p(s_{1:T})}\left[\mathbb{D}_{\mathrm{KL}}\Big(p(z_{1:T}|s_{1:T})||p(z_{1:T})\Big)\right]}_{\geq 0}$$

$$- \underbrace{\sum_{t=1}^{T-1} \mathbb{E}_{p(s_{t+1}, a_{1:t}, z_{1:t})}\left[\mathbb{D}_{\mathrm{KL}}\Big(f(z_{t+1}|s_{t+1})||q(z_{t+1}|z_t, a_t)\Big)\right]}_{\geq 0}.$$

The second term, may in general be smaller than the first term, for example, when the variational distribution perfectly matches the encoder, and, thus, the second term does not upper-bound the mutual information $I(s_{1:T}, z_{1:T})$.

We can, however, derive RPC based on our total correlation perspective by using the variational distribution

$$q'(z_{1:T}, a_{1:T-1}) = p(z_1)p(a_1) \prod_{t=1}^{T-1} q_\eta(z_{t+1}|z_t, a_t)p(a_{t+1}) \tag{13}$$

instead of Eq 9.

### A.3 UPDATING Q FUNCTION

Following the standard recursive Bellman equation, the Q function with parameters $\upsilon$ can be optimized by minimizing the loss

$$L(\upsilon) = \mathbb{E}_{\mathcal{D}, f, \pi}\left[\left(Q_\upsilon(s_t, a_t) - y(s_t, a_t)\right)^2\right] \tag{14}$$

where the target is given by

$$y(s_t, a_t) = r^*(s_t, a_t, s_{t+1}) + \gamma(1-d)\left[Q_\upsilon(s_{t+1}, a_{t+1}) - \beta \log(\pi_\phi(a_t|s_{t+1}))\right] \tag{15}$$

with discounted factor $\gamma$ and termination flag $d$ and next action $a_{t+1}$ sampled from the current policy. We employ the independent target Q function to computer the target and stop the gradient through the target Q function.

## B EXPERIMENTAL DETAILS

### B.1 TASK SPECIFICATION

We test our algorithms on MuJoCo-powered continuous control tasks from the Deepmind Control, which provides a standardized set of benchmarks for reinforcement learning agents. For each task, the

episode length is set to 1000 steps, and the action vector is bounded into $[-1, 1]$. We refer to (Tassa et al., 2018) for more descriptions of tasks.

## B.2 Implementation Details

**SAC codebase.** We implement our algorithm on top of the common PyTorch implementation of the SAC algorithm (Yarats et al., 2021). We used the default hyperparameters from that implementation unless specified otherwise. Detailed descriptions of the SAC implementation are available in (Yarats et al., 2021).

**Encoder.** The encoder $f_\theta(z_t|s_t)$ is parametrized as a 3-layer neural network with FCN (units=256) $\to$ FCN (units=256) $\to$ FCN (units=60) architecture and ReLU hidden activations. Its output is divided into the mean and the standard deviation of a diagonal Gaussian distribution.

**Prediction models.** Our prediction models $q_\eta(z_{t+1}|z_{1:t}, a_{1:t})$ and $q_\eta(a_t|z_{1:t}, a_{1:t-1})$ are parameterized by an LSTM module followed by a 3-layer neural network. The LSTM module is implemented using the common nn.LSTM class provided by PyTorch. The hidden dimension is set to 256, the output dimension is set to 30, and the number of recurrent layers is set to 1 for the LSTM module. The 3-layer neural network has the same architecture and activation function as the encoder. The output of the dynamic model is normalized and then divided into the mean and the standard deviation of a diagonal Gaussian distribution.

**Dual multipliers.** We treat the hyperparameter $\alpha$ as a dual multiplier and optimize it via dual gradient ascent. Following common practice (Haarnoja et al., 2018; Eysenbach et al., 2021), we initialize the value of $\alpha$ to $10^{-6}$ and parametrize it as $\log \alpha$ to ensure that it remains positive during optimization. For optimizing the entropy coefficient, we take the contribution from $\alpha$ into account and directly optimize $\beta' = \beta + \alpha$.

## B.3 Other Hyperparameters

We initialize the replay buffer with 5000 samples from the initial policy and train all agents for 1 million steps. We evaluate the agent every 20000 steps. All learnable parameters are updated using the Adam optimizer with a learning rate of $10^{-4}$. We determine information constraints $I_p$ and history length by performing hyperparameter tuning. We provide an overview of our used hyperparameters in Table. 2. For other details, please refer to the provided code.

Table 2: Hyperparameters used in MTC.

| Parameter | Value |
|---|---|
| information constraint $I_p$ | -3.0 |
| history length | 15 |
| Replay buffer capacity | 1 000 000 |
| Optimizer | Adam |
| Critic learning rate | $10^{-4}$ |
| Critic Q-function soft-update rate | 0.01 |
| Critic target update frequency | 2 |
| Actor learning rate | $10^{-4}$ |
| Actor update frequency | 1 |
| Actor log stddev bounds | [-10 2] |
| Temperature learning rate | $10^{-4}$ |
| Initial temperature | 0.1 |
| Initial steps | 5000 |
| Discount | 0.99 |
| Initial $\alpha$ | $10^{-6}$ |
| $\alpha$ learning rate | $10^{-4}$ |
| Representation dimension | 30 |
| Number of training steps | $10^6$ |
| Batch Size | 256 |

### B.4 EXTENDED DESCRIPTION OF BASELINE IMPLEMENTATIONS

**SAC.** We obtain the results for SAC by running the PyTorch implementations of SAC (Yarats et al., 2021). We use the same hyperparameters for SAC as our algorithm to ensure a fair comparison. We found that our obtained results for SAC are stronger than the results of SAC reported in previous work (Yarats & Kostrikov, 2020).

**LZ-SAC.** We use the official implementation provided by Saanum et al. (2023) to obtain the results for LZ-SAC, since the official implementation is based on the same codebase of SAC and the hyperparameters has been tuned to achieve good results on DMC tasks.

**RPC.** To obtain the results for RPC, we first use the original code provided by Eysenbach et al. (2021), which is built on top of the SAC implementation from TF-Agents. To achieve as good performance as possible for RPC, we perform hyperparameter tuning to select the suitable information constraint for RPC. To ensure a fair comparison, we additionally implement RPC by ourselves, using the same codebase of SAC as MTC and LZ-SAC. We use the same SAC hyperparameters for our implementation of RPC as our algorithm.

### B.5 COMPUTE RESSOURCES

We performed every experiment on an Intel(R) Xeon(R) E5-2620 CPU with GeForce GTX 2080 Ti graphics card and used approximately one day for training.

### B.6 ROBUSTNESS TO OBSERVATION NOISE

In all experiments of robustness and trajectory compression, for each agent, we evaluated the performance of policies saved after finishing the training for 1M steps. Gaussian noise is regarded as a strong state distractor for reinforcement learning algorithms in prior work (Bai et al., 2021). We add the Gaussian noise to observations and the learned policies select the actions based on the noisy observations.

### B.7 ROBUSTNESS TO ACTION NOISE

Noise added to actions can be viewed as a type of environment perturbation. In this experiment, we first use the saved policies to select the action based on the current state, and add the Gaussian noise to the chosen action. We then clip the action into $[-1, 1]$ before passing the action signal to the task.

### B.8 ROBUSTNESS TO FRICTIONS

Modifying the friction of the robot body to test the robustness of learned policies has been investigated in previous work. In our experiment setup, we get the body friction of the robot via the env.physics.model.geom_friction attribute provided by the environment. Since the body friction varies across different tasks, we change the frictions by scaling it, rather than increasing or decreasing a constant. We then evaluate the performance of the learned policies on the environment with the changed frictions.

### B.9 TRAJECTORY COMPRESSION

In our experiment, we measure the compressibility of trajectories using the bzip2 algorithm, which is easily available by installing the common bz2 python package. For each seed, we collect 30 trajectories using learned policies. Since the collected trajectories have the same number of data points and these data points have the same numerical precision, the uncompressed trajectories collected by the different algorithms have the same file size. We compress individual trajectories by calling the bz2.compress() method provided by the bz2 package. Smaller file sizes of compressed trajectories mean that trajectories can be better compressed.

### B.10 Hyperparameter Ablation

In our hyperparameter ablation experiment, we train our algorithm with each $I_p$ for 1 million steps on the Walk Stand task. The performance is evaluated by computing the average rewards over 10 episodes every 20000 steps. We save the learned policies after finishing training at 1M steps. Using the same experimental setup as before, we evaluate the performance of MTC for difference $I_p$.

## C Additional results

### C.1 Visualizations of Trajectories

For the Finger Spin task, the dimensions of the action and state space are 2 and 9, respectively. Fig 5 and Fig 6 visualize the action and state trajectories produced by our method and baselines on the Finger Spin task. We observed that MTC produces more consistent and periodic patterns in trajectories.

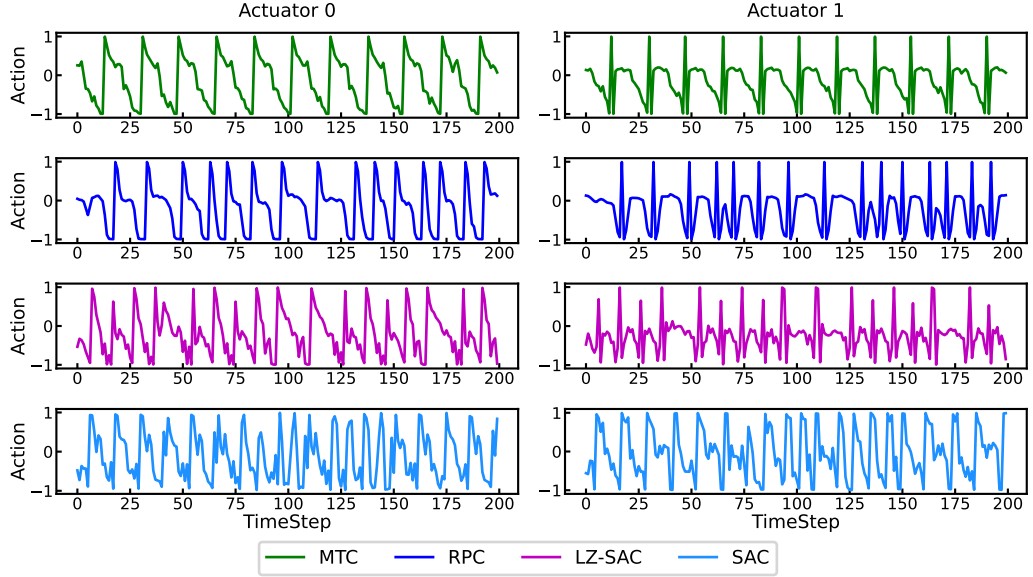

Figure 5: Visualizations of action sequences generated by our method and baselines on the Finger Spin task. MTC produces more consistent and periodic behavior than baselines.

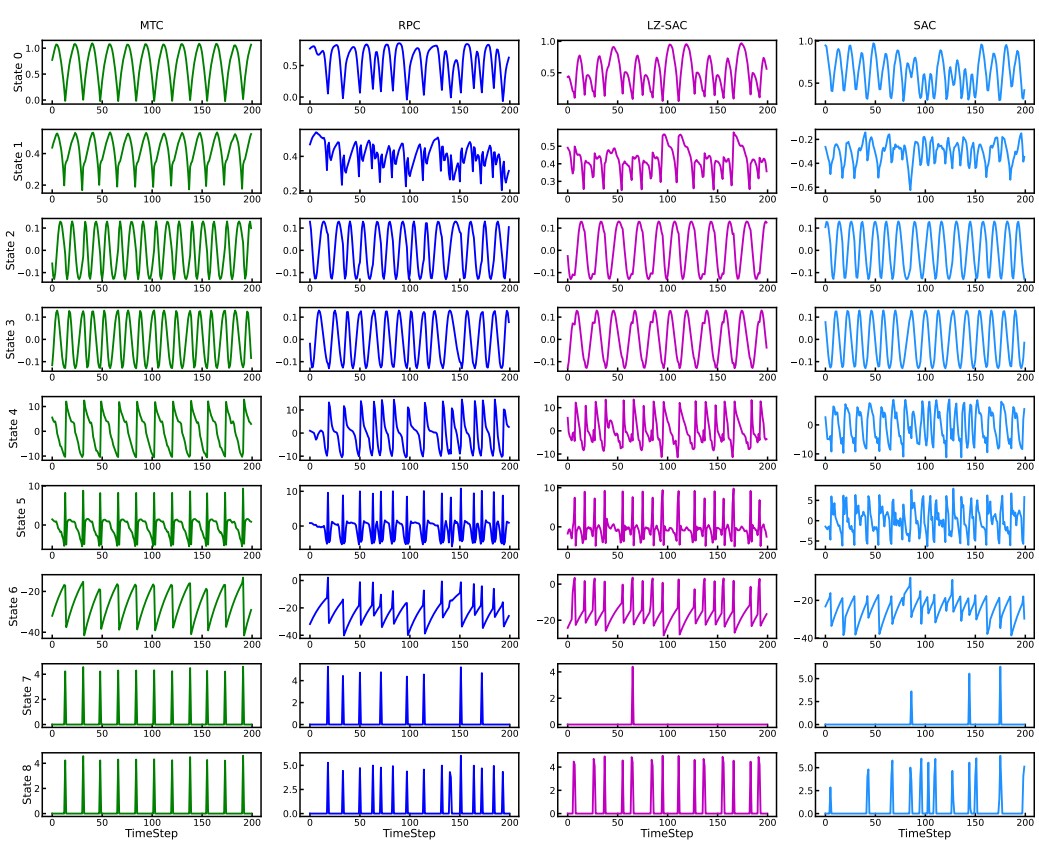

Figure 6: Visualizations of state sequences generated by our method and baselines on the Finger Spin task. State sequences of our method show more repeating and periodic patterns.

