# OpenReview forum: "Maximum Total Correlation Reinforcement Learning"
_ICLR.cc/2025/Conference — Submitted to ICLR 2025_

### Official Review · Reviewer_DJ5C · 2024-10-27

**Soundness:** 3
**Presentation:** 3
**Contribution:** 2
**Rating:** 5
**Confidence:** 4

**Summary:**

This paper proposes regularizing standard reinforcement learning (RL) with total correlation between state representations and actions in the whole trajectories to achieve *simpler* and more robust policies.

**Strengths:**

- The motivation and presentation are clear and easy to follow.
- The experimental results effectively demonstrate that the method learns periodic policies.

**Weaknesses:**

- The concept of total correlation is quite similar to mutual information, which has been extensively explored in both single-agent and multi-agent RL. Therefore, the novelty is limited, with the primary distinction being that prior work considers state-action pairs, while total correlation extends this to entire trajectories.
- Although the method was evaluated on 8 tasks in the DeepMind Control (DMC) suite, these tasks are relatively simple and inherently periodic. For such tasks, structurally designed policies or dynamic movement primitives (DMP) could potentially be more efficient. Therefore, it is essential to assess the generalizability of maximizing total correlation in more complex tasks, such as robotic grasping or kitchen environments.

**Questions:**

See the Weaknesses.

---

### Official Review · Reviewer_dKjy · 2024-10-29

**Soundness:** 2
**Presentation:** 3
**Contribution:** 2
**Rating:** 5
**Confidence:** 4

**Summary:**

This work proposes a novel RL method called MTC-RL, which introduces the total correlation as a key inductive bias for learning robust and generalizable policy and consistent state representation. The proposed method encourages the policy and state encoder to generate more simple and consistent behaviors and representations by maximizing the total correlation within the induced trajectories. In other words, **the authors claim that simultaneously learning state-consistent representation and action-consistent policy 1) makes RL algorithms more general and robust, 2) can save the amount of information of all states and actions within trajectories, and 3) produces improvements in final RL performance.** The experimental section demonstrates various supports to support the claims, resulting in better robustness to noise and changes in dynamics, particularly in robot control tasks.

**Strengths:**

1. The claims 1), 2), and 3) are partly supported by experiments in Sections 5.2, 5.3, and 5.1, respectively, providing a proper presentation of this work.
2. This work provides a derivation using the lower bound of the total correlation, provides mathematical support, and produces a unified objective containing two information regularization techniques: state-consistency and action-consistency.

**Weaknesses:**

**The main claims are not sufficiently supported.**
1. While this paper emphasizes the unified framework of state-consistency and action-consistency, the authors did not investigate whether the unified framework is required for their results. I strongly recommend conducting an ablation study using either state-consistency or action-consistency to support claims 1), 2), and 3).
2. Regarding claims 1) and 2), although this paper includes the state-of-the-art algorithm LZ-SAC, that paper also includes another algorithm called SPAC that imposes simplicity on action sequences. LZ-SAC paper reports that while LZ-SAC achieves better final performance than SPAC, SPAC shows better results in robustness and information efficiency. To fully support the claims 1) and 2), I recommend including SPAC as an additional baseline.
3. Regarding claim 1), while both RPC and LZ-SAC are expected to show robustness to dynamics changes, Figure 2 shows that they perform worse than SAC, which gives inconsistency of experiments.
4. Some works such as SPAC use transformers for their action sequence compressor, while other works such as this work use LSTM. Could the authors show that the merit of their work comes from the proposed unified framework, not the choice of the type of compressors?

**Questions:**

Please see the Weaknesses above. Here are additional questions:

1. While the modified reward (eq.4) could explain how it biases consistent behaviors and latent representations, it is unclear how maximizing the total correlation in the original objective (eq.1) biases the policy and encoder towards such behaviors and representations. Could the authors provide an explanation or an intuitive toy example?

2. Are there any experimental results in environments with higher dimensions? For example, tasks where the input is an observation instead of a state.

---

### Official Review · Reviewer_DvYG · 2024-10-30

**Soundness:** 2
**Presentation:** 4
**Contribution:** 2
**Rating:** 5
**Confidence:** 2

**Summary:**

This paper proposes a method that learns a policy that produces compressible trajectories, or more precisely trajectories that have a high total correlation. The motivation is that a simple policy is probably more robust to spurious correlations, noise, etc. than a more complex policy.

The authors propose to achieve this by maximizing a variational lower bound of the total correlation, which practically result in training a dynamics model and an action prediction model along with the policy.
Results show that this works pretty well in experiments on DMC, and improves robustness to observation noise, action noise and changed friction.

**Strengths:**

* Training more robust policies is an important challenge for RL and framing this through trajectory compressiblity is a very natural and nice formulation.
 * The paper is very well written and easy to follow
 * Experiments show that the method does indeed result in more periodic, more compressible trajectories and is more robust to noise.

**Structured into claims**:
 * Claim 1: Proposed method generates periodic/compressible trajectories. *Evidence*: Visualizations of actions over a trajectory indeed look very periodic. Compressiblity is validated by using lossless compression methods.
 * Claim 2: Perioidicity improves robustness of policy. *Evidence*: Policies are indeed more robust to observation noise, action noise and changed friction coefficient.

**Weaknesses:**

* My primary concern is the definition of the regularized reward function $r^*(s_t,a_t,s_{t+1})$. This reward depends on the entire history of the episode through the dynamics model $q_\mu(z_{t+1}|z_{1:t},a_{1:t})$. It should thus correctly be written $r^*(s_{a:t},a_{1:t},s_{t+1})$ and is no longer Markovian. However, this non-markovianness or its implications are not discussed anywhere. I would expect this to cause severe instabilities in training, but maybe that's not the case? At the very least this should be discussed and I would like to see some visualization of the training dynamics in the form of a training step vs reward plot.

 * The experimental validation is somewhat limited. Hyperparameter optimization is not performed in a structured way, both for the baselines and the proposed method.
 * The introduction motivates the method through combatting spurious correlation but there are not distractors etc. present in the experiments that could lead to spurious correlation beyond a noisy state observation. Likewise, the utilization of a learned state representation $z$ is motivated by the noisy TV problem, but this also isn't directly investigated in the experiments.
 * An ablation of only using the latent-dynamics model but not the action model would also be useful to see the contribution of the different parts of the method. This would be somewhere between the proposed method and RPC and would thus be very interesting to see.
 * This method adds additional dynamic models that are trained along with the the policy, so naturally the training time should be longer.

**Structured into claims:**
 * Claim 3: The method reduces spurious correlation and focuses only task-relevant state information by using a learned state encoding. *Evidence*: Only indirect evidence is given through experiments with observation noise. I don't think this is sufficient to support the claim and experiments with explicit distractors would be more convincing.
 * Claim 4: Maximum Total Corrlation SAC as a well motivated RL method. *Evidence*: The derivation hides the non-markovianness/nonstationarity of the reward function by incorrectly listing its arguments. This is my main concern with this paper and not discussed at all. The experiments indicate that it's practically not a big issue but I would like this to be elucidated more.



Minor comments:
 * Line 278: missing closing bracket for expectation
 * Line 671: $q(a_{t+1}...$ should be $q_\mathcal{X}(a_{t+1}...$
 * It would also be nice to discuss the connection to works that directly learn periodic controllers such as [1], and maybe even the connection to periodic locomotion controllers which are popular in classical robotics.

[1] Raffin, Antonin, et al. "An Open-Loop Baseline for Reinforcement Learning Locomotion Tasks." RLC 2024

**Questions:**

* I would apprecite if you could reply to my concern above about the non-markovianness regularized reward function.
 * As the regularized reward could affect training stability, could you show some training-step vs reward training curves?
 * As you are training additional models, the training probably takes significantly longer than normal SAC and even RPC. Could you show a wall-time comparison?

Overall I really like the idea of the paper and the proposed method, but I do have the listed concerns about the regularized reward and about the limitations of the experimental validation.

I'm willing to raise my score if the concerns are addressed.

---

### Official Review · Reviewer_gUZZ · 2024-11-03

**Soundness:** 2
**Presentation:** 3
**Contribution:** 1
**Rating:** 3
**Confidence:** 4

**Summary:**

The paper proposes an auxiliary RL objective, MTC, that maximizes the total correlation of states and actions under the policy. This objective encourages the policy to learn simple, consistent, and compressible behaviors that are robust to noise and dynamics shifts. Specifically, the authors assume a latent-state-conditioned policy and incentivize the policy to maximize the total correlation of latent states and actions: $\mathcal C(z_1; a_1; \dots; a_{T-1}; z_T)$. This objective is intractable, so they estimate a variational lowerbound $\mathbb E_{\pi, f} \left[ \sum_{t=1}^{T-1} \left[\log \frac{q_{\eta}(z_{t+1}|z_{1:t}, a_{1:t})}{f_{\theta}(z_{t+1}|s_{t+1})} + \log\frac{q_\chi(a_t|z_{1:t}, a_{1:t-1})}{\pi_\phi(a_t|s_t)}\right]\right]$ using a dynamics model $q_\eta$, an action prediction model $q_\chi$, and an encoder $f_\theta$. The estimated lower bound is added to the reward at each timestep, with its weights dynamically adjusted via constrained optimization. Experiments show that MTC combined with SAC outperforms baselines on a suite of DM Control tasks. They further show that MTC is more robust than baselines in the presence of state and action noises and dynamics shifts.

**Strengths:**

1. Maximizing the total correlation is a reasonable objective to induce simplicity in RL policies. The total correlation measures the KL divergence between the independent state and action distributions and their joint distribution. The higher the total correlation, the less it costs to compress the entire trajectory compared to compressing the states and actions independently.
2. The authors propose a variational lower bound of the total correlation. The mathematical derivations are correct and easy to follow.
3. The ablation experiment in Section 5.4 and Figure 4 showing improved generalization with a higher target total correlation value lends credibility to MTC's effectiveness.

**List of Claims**

Claim 1: the MTC objective improves RL agents' robustness to noise and distribution shifts.
- Assumption: simplicity leads to robustness.
- Evidence: ablation in Section 5.4 and Figure 4.
- Statement: this ablation experiment shows that as they increase the target total correlation (the target constraint value in the constrained optimization problem), the method becomes more robust to observation and action noises and friction changes.

**Weaknesses:**

1. The fundamental motivation of the paper, "simplicity is a powerful inductive bias", is not well supported. The authors support it by enumerating examples in related problems, e.g. "regularization is used for simpler policies, data augmentation for simpler representations, and sparse reward functions for simpler objectives." But this is quite vague. It might very well be that in RL, overoptimizing for simplicity leads to suboptimality in complex tasks.
2. In Section 4.1, the authors motivate the use of latent states by stating that it is more robust to distractions from task-irrelevant factors (e.g. noisy TV). This claim again lacks support.
3. The key differences between MTC and RPC [1] are (1) MTC considers states and actions while RPC only considers states, and (2) MTC conditions the dynamics and action prediction models on the history, whereas RPC only conditions on the current state. I'm not sure why (1) would conceptually improve upon RPC (see question below). (2) introduces a confounding variable that is orthogonal to the claim, since we can replace the model in RPC with a recurrent model and hope to see better performance.
4. While the goal is to induce simplicity in the policy's behavior, the method requires learning three additional models to estimate an auxiliary objective, which is quite cumbersome.
5. The rest of experiments are unconvincing. While the main results (Table 1) show MTC performing on par or slightly better than baselines, the improvement is not significant. Moreover, these results are in a narrow domain that prefers repetitive motion (e.g. Hopper, Cheetah, Walker are all locomotion tasks). I'm not sure if the same holds for more complex tasks. The robustness results in Figure 2 don't imply MTC being *more* robust than baselines, because SAC shows the same growth trend, just starting from a lower initial value. What I hope to see is that MTC performance degrades *at a slower rate than* SAC, instead of having a constant offset.

Overall, I think this paper can be significantly strengthened by (1) grounding it in a concrete problem setting like sim2real transfer instead of a lofty desideratum (i.e. simplicity), and (2) either removing the unwarranted claims or adding experiments to support them. I strongly recommend the authors to revise and resubmit.

**List of Claims (cont'd)**

Claim 2: simplicity is desirable for RL.
- Assumption: none
- Evidence: slightly better empirical performance of MTC over baselines on DM control tasks.
- Statement: I don't think there is enough theoretical or empirical evidence to support this claim. Given that the DM control tasks prefer periodic behavior, it might be that simplicity is particularly useful in this domain.

Claim 3: using latent states results in less distraction from task-irrelevant factors.
- Assumption: the environment consists of task-irrelevant distractors, modeling which hinder the policy's performance.
- Evidence: none
- Statement: to support this claim, the authors need to compare their method to a baseline that uses raw observations on environments with distractors.

Claim 4: MTC improves upon RPC by considering states and actions, and conditioning the models on the history.
- Assumption: none
- Evidence: slightly better empirical performance than RPC.
- Statement: I'm not sure "why" these changes would improve upon RPC. The sequence of states should determine the sequence of actions to a large extent. Conditioning on the history is a choice of variational approximation that can also be applied to RPC, and the paper doesn't compare to RPC + recurrent models.

[1] Benjamin Eysenbach, Ruslan Salakhutdinov, Sergey Levine. Robust Predictable Control. NeurIPS 2021.

**Questions:**

1. Why is simplicity a good inductive bias for RL? Does this inductive bias hold outside of locomotion domains where repetitive motion is preferred?
2. Why is accounting for the states and actions better than only accounting for states as in RPC? Isn't the state trajectory a consequence of the action trajectory?
3. In Equation 2, what does it mean to take expectation under $f$?
4. Does the infinite horizon variant correspond to a meaningful information-theoretic objective?
5. Can you provide more experiments to support your claims?
    - a manipulation task where the motion is less periodic, to support the claim that simplicity is a universally desirable property.
    - visual experiments with distractors (e.g. distracted DM control) to support the claim that latent-state policies are robust to distractors.
    - visual robustness experiments with lighting / texture change similar to the sim2real setting, to support the claim that MTC is robust to distribution shifts.

---

### Meta-Review · Area_Chair_65ot · 2024-12-20

**Metareview:**

This paper proposes a method to learn policies that are robust to noise and changes in dynamics. The method works by adding a new term to the objective, encouraging the agent to find a policy that is highly rewarding and produces consistent trajectories. The reviewers all suggest that the primary claims of the paper were not sufficiently supported. There was no response from the authors. Thus, I am recommending that this paper be rejected.

**Additional Comments On Reviewer Discussion:**

There was no response from the authors.

---

### Decision · Program_Chairs · 2025-01-22

Reject